# Landscape of Phenotype-Genotype Correlations in Romanian Patients with Medullary Thyroid Carcinoma

**DOI:** 10.3390/cancers18010093

**Published:** 2025-12-27

**Authors:** Laura-Semonia Stanescu, Sofia-Maria Lider-Burciulescu, Andrei Muresan, Sorina Violeta Schipor, Elena Braha, Monica Livia Gheorghiu, Corin Badiu

**Affiliations:** 1Department of Endocrinology, “Carol Davila” University of Medicine and Pharmacy, 0505474 Bucharest, Romania; laura-semonia.stanescu@drd.umfcd.ro (L.-S.S.); corin.badiu@umfcd.ro (C.B.); 2Department of Physiology, “Carol Davila” University of Medicine and Pharmacy, 0505474 Bucharest, Romania; sofia.timpuriu@umfcd.ro; 3”Ana Aslan”, National Institute of Geriatrics and Gerontology, 011241 Bucharest, Romania; 4Department of Research, “CI Parhon” National Institute of Endocrinology, 011863 Bucharest, Romania; andrei.muresan@parhon.ro (A.M.); sorina.schipor@parhon.ro (S.V.S.); elena.braha@parhon.ro (E.B.)

**Keywords:** medullary thyroid cancer, RET, sporadic, hereditary, mutations, germline, somatic

## Abstract

The aim of the research was to characterize RET mutations in patients with medullary thyroid carcinoma (MTC) in Romania, considering their relevance for tyrosine kinase inhibitor therapy. Up until this study, due to recent restrictions, no other center in Romania had specifically tested for somatic RET mutations to enable access to targeted therapy. This study included both somatic and germline testing for RET mutations, providing novel insights into the mutational landscape of the Romanian population.

## 1. Introduction

MTC is a neuroendocrine malignant tumor that originates from parafollicular C-cells of the thyroid gland. These cells secrete calcitonin (CTN) and carcinoembryonic antigen (CEA), which are used as biomarkers for the diagnosis and follow-up of MTC, along with other mediators (such as adrenocorticotropic hormone, ACTH) that cause paraneoplastic syndromes [1,2]. MTC is a rare tumor; it accounts for 0.4–1.4% of all thyroid nodules, about 2% of all thyroid cancers, and about 0.14% of all thyroid cases in subjects who underwent autopsy [3,4,5]. MTC exhibits a wide range of clinical behaviors, from indolent to aggressive. The survival rate depends on the stage at diagnosis; thus, the 5-year relative survival for stages I to III is 93%, while in stage IV, it drops significantly to 28% [1,6].

In 1985, Takahashi and colleagues [7] discovered the proto-oncogene RET in a fibroblast cell line using DNA from human lymphoma cells. Since then, important progress has been made in understanding its role in thyroid cancer. In 1993, single amino acid substitutions, small insertions, and deletions in the RET gene were identified in patients with hereditary MTC [8]. The RET proto-oncogene encodes a transmembrane tyrosine kinase receptor that is expressed in thyroid C-cells [9]. Ligand binding induces RET dimerization, leading to autophosphorylation of the intracellular kinase domain. Subsequently, specific intracellular pathways are activated, which are involved in cell proliferation, differentiation, survival, and growth [10,11].

MTC can be hereditary (25% of cases) or sporadic (75% of cases). Hereditary MTC occurs as a monogenic autosomal dominant disorder in multiple endocrine neoplasia types 2 and 3 (MEN2 formerly MEN2A, MEN3 formerly MEN2B, according to the latest WHO classification of neuroendocrine neoplasms) [12].

Figure 1 describes the clinical characteristics of hereditary MTC, which includes the four subtypes of MEN2 syndrome and MEN3 syndrome. The most frequent variant is classic MEN2, which accounts for 80% of hereditary MTC cases and features MTC, pheochromocytoma (PHEO), and primary hyperparathyroidism (HPTH). Other variants of MEN2 include MEN2 with cutaneous lichen amyloidosis (CLA), MEN2 with Hirschsprung disease (HD), and familial medullary thyroid carcinoma (FMTC). MEN3 syndrome is rare (5% of hereditary MTC) and has a distinct clinical presentation: MTC, PHEO, generalized ganglioneuromatosis, ocular abnormalities, and a particular phenotype (typical facies, skeletal malformations, and marfanoid habitus) [1,13,14,15].

RET mutations exhibit distinct biological behaviors that underlie the specific clinical phenotypes. In the majority of patients with MEN2 syndrome (approximately 95%), germline RET mutations are located within the extracellular cysteine-rich domain (CRD); the affected codons are 609, 611, 618, and 620 (exon 10) or 630 and 634 (exon 11), with mutations in codon 634 being the most frequent [13,16,17]. In particular, in FMTC, the most common RET mutation affects extracellular cysteines other than 634 or intracellular codons 768, 790, or 804 (exons 13 or 14), with the codon 804 mutation being the most frequent [13,16,17,18].

MEN3 is associated with RET mutations in the intracellular tyrosine kinase domain, predominantly the M918T mutation in exon 16 [19,20]. It has also been linked to the A883F mutation or double mutations involving V804M [21,22].

In all hereditary forms, germline RET screening should be applied to first-degree relatives to enable personalized surveillance for early diagnosis and treatment, thereby improving patient survival.

In sMTC, somatic mutations are present in up to 55% of cases [23]. The most common somatic RET mutation is M918T (40% of cases), which is associated with the highest aggressiveness [1,23]. Other somatic RET mutations are rare and have been described in codons 611, 618, 620, 630, 634, 768, A883, and 891 [23,24]. In Romania, the detection of a somatic pathogenic or likely pathogenic RET mutation is required in advanced or metastatic disease to qualify for reimbursed systemic therapy. Characterization of genetic alterations in tumor cells plays an important role in determining the best treatment strategy. For MTC, RET mutational status is critical for selecting optimal targeted therapy. Advances in next-generation sequencing (NGS) have identified novel oncogenic alterations in RET-naive MTCs, revealing potential therapeutic targets.

However, traditional Sanger sequencing offers some important advantages specific to patient samples collected. NGS is the gold standard in clinical laboratories and for regulatory approval (FDA, CLIA) due to its precision in sequencing targeted regions, such as confirming specific mutations or individual genes. Sanger sequencing is highly accurate for sequencing short DNA fragments (<1000 base pairs) with low error rates [25,26]. In addition to the apparent simplified workflow and cost-effectiveness inherent to the Sanger sequencing technique, it provides targeted sequencing of low-quantity DNA [26] or degraded samples, as its procedure involves PCR to amplify specific regions. NGS requires higher DNA input and quality, which is not feasible for small biopsies or certain sample types like Formalin-Fixed Paraffin-Embedded (FFPE) [27].

In this study, we aimed to characterize germline RET alterations in our probands and, to a limited extent, their families (*n* = 32) using Sanger sequencing. In cases of progressive metastatic disease, we aimed to identify targetable somatic mutations via sequencing to guide systemic therapy.

## 2. Materials and Methods

### 2.1. Patients

The study was conducted between 2021 and 2024, with patient recruitment and data collection commencing only after ethical approval was obtained from the Ethics Committee and Scientific Council of the “C.I. Parhon” National Institute of Endocrinology, Bucharest, Romania (approval number: no. 12, date of approval: 27 June 2022) Germline RET testing was performed in 164 Romanian participants: 20 with clinically diagnosed hereditary disease, 112 with apparently sporadic MTC, and 32 first-degree relatives. Somatic RET mutations were evaluated in 27 wild-type cases of aggressive disease. Sporadic MTC was defined as the absence of a familial history of MTC and no personal history of other disorders possibly related to RET alterations. Informed consent for RET genetic testing and use of clinical and laboratory data was obtained from all participants.

### 2.2. RET Mutation Genetic Analysis

#### 2.2.1. Germline RET Mutation Testing

Genomic DNA was purified from peripheral blood-EDTA using the Wizard Genomic DNA Purification Kit (Promega, A1620, Madison, WI, USA). To verify the quantity and quality of the extracted DNA, we used spectrophotometry (DeNovix DS-11, Wilmington, DE, USA) and fluorimetry using the Qubit 2.0 Fluorometer (ThermoFisher Scientific, Q32866, Carlsbad, CA, USA) with a DNA-specific dye using Qubit™ dsDNA Quantification HS Assay Kit (ThermoFisher Scientific, Q32851, Carlsbad, CA, USA).

All patients underwent germline RET testing for MTC, according to international guidelines. Relatives were screened specifically for the familial RET variant, whereas probands were analyzed across eight exons (11, 10, 14, 16, 13, 15, 8, and 5) in order of reported mutation frequency.

RET mutations were identified in the Molecular Endocrinology Laboratory of “C.I. Parhon” National Institute of Endocrinology and “Carol Davila” University of Medicine and Pharmacy. Exons were amplified using conventional PCR and verified on 2% agarose gels. PCR products were purified using Ampure, sequenced using fluorescent terminators, and cleaned with CleanSeq before loading onto the sequencer. Sequences were aligned to the hg38 reference genome (UCSC BLAT) and analyzed via chromatogram comparison using the software Sequence Investigator that comes with the Beckman CEQ™ 8000 Genetic Analysis System (Beckman Coulter, Fullerton, CA, USA).

#### 2.2.2. Somatic RET Mutation Testing

Germline RET wild-type cases were selected for somatic RET mutation analysis; specifically, exons 16 and 11 were tested for. DNA was obtained from fresh tissue after thyroid surgery in 10/27 cases or retroactively from FFPE tissue blocks collected from primary (15/27 cases) or metastatic tumors (2/27 cases). DNA extraction from FFPE samples was carried out using the Analytik Jena Black PREP FFPE FNA kit (AJ Innuscreen GmbH, 845-BP-0021250, Berlin, Germany) and DNA extraction from fresh tissue was carried out using the Purelink Genomic DNA mini kit (Invitrogen, K182002, Carlsbad, CA, USA). Samples with insufficient DNA yields were excluded from the analysis.

### 2.3. Statistical Analysis

A chi-square test (df = 1, alpha = 0.05) was performed to determine the significance of the cys variant (or non-cys variants) when compared to the total prevalence in patients. A two-tailed independent *t*-test (alpha = 0.05) was performed to determine whether there was a significant difference in the mean age between patients with germline RET variants (59) and wild-type patients (105).

## 3. Results

In total, we performed RET genetic testing for germline mutations in 164 subjects. Of these, 112 patients had apparently sporadic forms of MTC (no family history of MTC, no other endocrine disease), and 52 patients had a positive history of disease, of which 20 were clinically affected by a hereditary disease at diagnosis, and 32 cases were relatives of RET-positive MTC patients. To provide a comprehensive overview of the study population, Table 1 outlines the demographic and clinical characteristics of patients with MTC.

A total of 59/164 (35.9%) patients were identified with the germline RET variant, of which 11/112 (9.8%) patients with apparently sporadic cases were reclassified as hereditary, 28/32 (87.5%) relatives of RET carriers, and all (20/20) patients had a family history of disorders related to RET alterations. Furthermore, the relatives of probands were invited to participate in genetic screening.

Eleven different RET variants were identified in our cohort, classified according to the current ATA guideline risk class as follows: five moderate (MOD) risk, five high (H) risk, and one of the highest (HST) risk level. The HST risk level identified was Met918Thr, which was found to be a de novo germline mutation [1].

Mutations affecting the cysteine at codon 634 (C634R/Y/S/F/W) were the most prevalent in our series, found in 33/59 (55.9%) cases. When considering clinically hereditary cases, C634W was the most frequent mutation in 14/48 cases (29.2%), in contrast to apparently sporadic cases, in which V804M was the most prevalent mutation.

When considering all mutations within the extracellular CRD, a significantly higher prevalence was observed in already known syndromic cases than in apparently sporadic cases (45/48, 93.7% vs. 6/11, 54.5%) (*p* = 0.006, chi-square stat = 11.755, alpha = 0.05). Germline mutations affecting non-cysteine residues were more frequent in apparently sporadic cases than in already known syndromic cases (5/11, 45.4% vs. 3/48, 6.25%) (*p* = 0.006, chi-square stat = 11.755, alpha = 0.05).

### RET Test Results and Clinical Correlations

The mean age at MTC diagnosis was 30 years (range 3–70) in patients with germline RET variants and 51 years (range 5–76) in wild-type patients (*p* = 0.00001, t-value = −7.90976, two-tailed independent *t*-test, alpha = 0.05).

Among all hereditary cases, we found 58/59 (98.3%) cases with MEN2 syndrome and 1/59 (1.7%) case with MEN3 syndrome. Of the MEN2 subtypes, we found 42/58 cases (72.4%) with classic MEN2, 3/58 cases (5.2%) with MEN2 with CLA, 3/58 cases (5.2%) with MEN2 with HD, and 10/58 cases (17.2%) with FMTC in our cohort. MTC diagnosis was the first component of MEN syndrome in 51 cases and was secondary in eight of the 59 cases.

The most frequent mutation in classic MEN2 cases was in codon 634 in 30/42 cases (71.4%), followed by mutations in codon 618 in 7/42 cases (16.7%) and in codon 620 in 5/42 cases (11.9%). The most frequent RET mutation in FMTC was in codon 804 (7/10 cases, 70%), while the other 3/10 cases (30%) had mutations in codon 620. All patients with HD were positive for RET mutations in exon 10 (one patient in codon 618 and two patients in codon 620). As shown in Table 2, among cases with the C634 mutation in exon 11, 16/29 (55.2%) had PHEO and 9 (31%) had HPTH.

Among patients with a mutation in exon 10, 4/18 (22.2%) had PHEO (all cases harboring a mutation in codon 618), while none had HPTH. In our series, none of the cases with a non-cysteine mutation had PHEO or HPTH. Among the cases with a germline RET mutation in codon 634, 3/29 (10.3%) had MEN2 with CLA. The case with MEN3 syndrome (M918T mutation) clinically presented with neurinomas of the tongue, lips, and buccal mucosa, and marfanoid habitus.

From the total of 105 sMTC in our series, we performed somatic RET testing on 27 cases using either FFPE or fresh tissue samples. We found that 20/27 (74%) cases were characterized by the presence of a pathogenic somatic mutation, while 7/27 (26%) cases were found to be negative.

As shown in Table 3, among the RET-positive tumors, the most prevalent somatic mutation was M918T in exon 16 (15/20 cases, 75%). In addition, we found three other somatic mutations: C634R mutation in exon 11 (2/20 cases, 10%), C634Y mutation in exon 11 (1/20 cases, 5%), and C630G mutation in exon 11 (1/20 cases, 5%) (for one patient, we have only information about the presence of the mutation in exon 11).

In our cohort, 18 patients with sMTC received multikinase tyrosine kinase inhibitor (TKI) therapy, specifically with vandetanib or cabozantinib. Among these 18 patients, 14 harbored somatic RET mutations, with the majority (10/14) carrying the M918T mutation.

## 4. Discussion

Genetic testing in patients diagnosed with MTC plays a key role in the development of individualized treatment with systemic therapies for cases with advanced disease.

Over the past decade, the National Institute of Endocrinology in Bucharest, Romania, has established the necessary infrastructure and developed standardized procedures for molecular diagnosis, genetic counseling, and targeted therapeutic interventions in MTC.

Literature reports that ~75% of MTC cases are sporadic and ~25% are hereditary, almost exclusively within MEN2/MEN3 syndromes [28]. In our Romanian cohort, sMTC comprised 105/164 (64.1%) cases, while hMTC comprised 35.9% of cases, a higher proportion than the global average. Germline RET mutations drive hMTC, with codon 634 mutations globally accounting for 36–50% of MEN2-associated mutations, predominantly in classic MEN2 (85% of MEN2 cases involve exon 11, especially codon 634) [1]. Codon 634 prevalence varies geographically: Western/Northern Europe reports 30–40% (France: 30.6% [29]; Italy: 34.8% [30]; Slovenia: 30% [31], whereas Southern cohorts exhibit higher rates due to founder effects (Turkey: 54.9% [32]; Spain: up to ~90% [33]. Our cohort revealed codon 634 in 55.9% of hMTC cases, exceeding the European average and aligning with the higher-prevalence Southern European cohorts. As expected, classic MEN2 was linked to mutations in the extracellular cysteine-rich domain, with codon 634 being the most frequent (71.4%), consistent with previous reports [34,35,36]. In contrast to the literature, where 10–15% of classic MEN2 cases harbor mutations in exon 10 (codons 609, 611, 618, and 620) [35], our cohort showed a markedly higher prevalence of 28.6%. Only one patient (1.7%) had MEN3, which, while underscoring the syndrome’s rarity, precludes reliable prevalence estimates due to the low number of cases.

In FMTC, germline mutations are known to be located throughout the RET gene, with a predominance in the extracellular cysteines other than codon 634 (618, 620–exon 10) and in the intracellular codons (768, 790, 791–exon 13 and 804, 844–exon 14), with mutations in codon 804 being the most frequent [34,35]. Our results are consistent with previous data, with 70% of FMTC cases being positive for mutations in codon 804.

The phenotypic characteristics associated with specific genetic alterations have been documented in the literature as follows: for mutations in codon 634, the clinical presentation features MTC in 100% of patients, PHEO in 50% of patients, and HPTH in 15% of patients; in addition, around 30% of patients have CLA, known as MEN2 with CLA [16,35]. Among our patients with mutations in codon 634, 55.2% had PHEO and 31% had HPTH. PHEO can also be found in 20% of patients with genetic alterations in exon 10 (codons 609, 611, 618, and 620) or exon 15 (codons 791 and 804) [37]. We also diagnosed PHEO in four cases of mutations in codon 618. In our Romanian cohort, all cases of HD had a mutation in exon 10 of the RET gene, consistent with the data reported in the literature [34].

Whereas classic MEN2 and MEN3 are clinically well-defined, the lack of specific clinical features and/or familial history makes the diagnosis of FMTC relatively difficult, thus generating an underestimation of FMTC prevalence [16].

Currently, RET genetic testing is considered the standard of care for defining the „hereditary” nature of MTC. Interestingly, in the Romanian cohort, genetic testing allowed the re-classification of 9.8% of MTC patients from sporadic to hereditary. This result is even higher than previous data, which ranged from 4% to 7.7% [38,39,40,41,42]. This disproportionately high fraction could be explained by selection bias, a limitation of this study. Because our cohort was recruited from a tertiary referral center specializing in endocrine genetics, the patient population was enriched with hereditary cases. This recruitment pattern limits the generalizability of our findings; therefore, the observed frequency should be interpreted with caution.

Genotype particularities of the disease have clinical benefits not only for index cases but also for gene carriers who can be identified because of this re-classification. According to the published literature, sMTC has been reported to exhibit a markedly higher prevalence of FMTC, accounting for approximately 87% of cases, in contrast to the distribution observed in classic MEN2 syndrome [43]. Our results showed a similar prevalence of FMTC (54.5%) and classic MEN2 syndromes (45.4%).

The findings of the present study are consistent with previous reports in the literature, which delineate distinct profiles of RET germline mutations in hereditary medullary thyroid carcinoma compared to cases that appear sporadic. Mutations involving cysteine residues were significantly more frequent in the group of clinically hereditary cases than in apparently sporadic cases. Conversely, mutations affecting non-cysteine residues are more frequent in apparently sporadic cases than in clinically observed familial cases. This difference can be explained by the higher oncogenic potential and penetrance of cysteine mutations compared to non-cysteine RET variants [44].

Sporadic disease has its peak incidence in the fifth decade of life, while hereditary disease is diagnosed earlier [35]. These observations are consistent with our findings, which showed that the mean age at MTC diagnosis was 30 years in patients with a germline RET mutation and 51 years in those with wild-type RET.

The most common somatic mutation described in the literature is M918T, found in almost 40% of patients with sMTC and associated with a poor prognosis [23]. In our cohort, 27 cases were tested for somatic RET mutations, of which 74% were positive. Consistent with previously published studies, the M918T mutation was the most prevalent, with a detection rate of 75%. Although other somatic mutations are rare, we found three other somatic mutations (C634R, C634Y, and C630G), all in exon 11.

Further characterization of the somatic mutational landscape in MTC tumor cells could provide valuable insights into tumor behavior and aggressiveness. Intratumoral genetic heterogeneity has been described in MTC, with distinct subclones harboring different mutations [16,23]. The emergence of additional somatic mutations may reflect clonal evolution during tumor progression [45]. Similarly, discordant RET mutational profiles between primary tumors and metastatic sites have been reported, suggesting the acquisition of secondary alterations during disease dissemination [46].

Somatic RET mutations, particularly the p.Met918Thr (M918T) variant, have been repeatedly associated with an unfavorable prognosis, including advanced tumor stage, higher T category, and increased rates of lymph node and distant metastases [23,47,48,49,50].

In our cohort, the 15 patients harboring somatic M918T mutations exhibited a consistently aggressive phenotype: the median age at diagnosis was 50 years (range 21–69); among the 13 patients with available staging data, 9 (69%) presented with stage IV disease; distant metastases were documented in five patients (33%), involving the liver (*n* = 3), lung (*n* = 3), and bone (*n* = 2); regional lymph node involvement was universal (15/15; 100%); and baseline serum calcitonin levels were markedly elevated (median 2500 pg/mL; range 250–24,600 pg/mL in 11 patients with quantifiable values, exceeding the upper limit of detection in the remaining 4). These observations align with prior reports of an aggressive clinical course conferred by the M918T alteration.

As for RET somatic mutations in MEN2-associated MTC, especially M918T, the impact on clinical course is still unknown; nevertheless, a somatic event is rarely looked for in MEN2 patients. In 1996, Marsh et al. [51] and Eng et al. [16] reported MEN2 cases with a C634 germline RET mutation and an additional M918T somatic mutation. However, the independent prognostic impact is difficult to isolate, given the variable penetrance of germline variants [1]. Lombardo et al. [52] published the case of a 12-year-old girl with invasive MTC harboring concomitant germline 804 and somatic M918T RET mutations, supporting the hypothesis that a somatic mutation could be responsible for an early clinical appearance and higher aggressiveness of the disease. Sahakian et al. [53] also reported a case of a 35-year-old woman with particularly aggressive MTC with double germline L790F and somatic M918T RET mutations. Taken together, these observations indicate that concomitant germline alterations (e.g., at codons 634, V804M, or L790F) and somatic M918T mutations may suggest that secondary somatic events contribute to increased disease aggressiveness or the manifestation of early-onset phenotypes.

A key limitation of the current study is that systematic somatic RET mutation testing was not performed in our hereditary cases, precluding direct assessment of such double-hit events in this subgroup. Therefore, further studies are warranted to elucidate the potential modifier roles of somatic alterations in MEN2-associated MTC.

## 5. Conclusions

In conclusion, this study provides the first comprehensive genetic characterization of MTC in a Romanian cohort, revealing a higher-than-expected hereditary fraction (35.9%) and highlighting the critical importance of germline RET screening in all MTC patients, irrespective of their clinical presentation or family history. Nearly 10% of patients initially classified as sporadic were reclassified as hereditary, demonstrating the value of a systematic molecular evaluation.

Codon 634 mutations dominated the germline landscape (55.9% of hereditary cases; 71.4% of classic MEN2), aligning with elevated prevalence patterns in Southern Europe. Notably, distinct mutational profiles emerged between known syndromic and apparently sporadic cases, with cysteine-rich domain mutations significantly enriched in the former and non-cysteine mutations in the latter. In advanced MTC, somatic M918T is the most common mutation (75%), strongly linked to aggressive disease, and guides tyrosine kinase inhibitor therapy.

Universal RET testing (including relatives screening) and targeted somatic mutation analysis in advanced disease are essential to enable risk-stratified surveillance, earlier prophylactic thyroidectomy, and timely personalized systemic therapy. This study supports population-adjusted management strategies and contributes novel data to the European MTC Genetic Atlas.

## Figures and Tables

**Figure 1 cancers-18-00093-f001:**
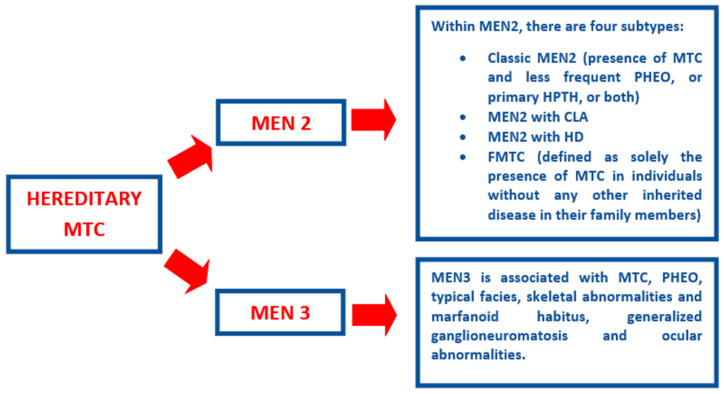
Characterizing Hereditary MTC in terms of MEN syndromes and their respective pathologies. Abbreviations: CLA, cutaneous lichen amyloidosis; FMTC, familial medullary thyroid carcinoma; HD, Hirschsprung disease; HPTH, hyperparathyroidism; MEN, multiple endocrine neoplasia; MTC, medullary thyroid carcinoma; PHEO, pheochromocytoma.

**Table 1 cancers-18-00093-t001:** Demographic and clinical characteristics of patients with medullary thyroid carcinoma.

Characteristics	Patients No. (%)
**Sex**	
Male	55 (33.5)
Female	109 (66.5)
**Age at MTC diagnosis (years)**	
Mean ± SD	44 ± 18
Median (IQR) Range	45 (31–58) 3–76
**Medullary thyroid carcinoma**	
Sporadic	105 (64.1)
Hereditary	59 (35.9)
**Hereditary MTC**	
**MEN2**	58 (98.3)
Classic MEN2	42 (72.4)
MEN2 with CLA	3 (5.2)
MEN2 with HD	3 (5.2)
FMTC	10 (17.2)
MEN3	1 (1.7)
**TNM stage MTC at diagnosis ***	
I	47 (42)
II	3 (2.7)
III	23 (20.5)
IV	39 (34.8)

* Patients with no data on the stage of MTC at diagnosis were not included in the table. Abbreviations: CLA, cutaneous lichen amyloidosis; FMTC, familial medullary thyroid carcinoma; HD, Hirschsprung disease; MEN, multiple endocrine neoplasia.

**Table 2 cancers-18-00093-t002:** Classified germline RET variants identified in the study cohort (*n* = 164; 112 apparently sporadic and 52 hereditary cases).

Exon	Nucleotide Variant	Aminoacidic Change	ATA Risk Level	No. Carriers ^a^	MTC/Tot Carriers (%) ^b^	PHPT/Tot Carriers (%)	PHEO/Tot Carriers (%)	CLA/Tot Carriers (%)	HD/Tot Carriers (%)	MEN3 Manifestations/Tot Carriers (%)
10	c.1858T>A	p.Cys620Ser	MOD	2	2/2 (100.0)	0/2 (0.0)	0/2 (0.0)	0/2 (0.0)	0/2 (0.0)	0/2 (0.0)
10	c.1858T>C	p.Cys620Arg	MOD	8	5/5(100.0)	0/8 (0.0)	0/8 (0.0)	0/8 (0.0)	2/8 (25.0)	0/8 (0.0)
14	c.2410G>A	p.Val804Met	MOD	6	5/5(100.0)	0/6 (0.0)	0/6 (0.0)	0/6 (0.0)	0/6 (0.0)	0/6 (0.0)
14	c.2410G>C	p.Val804Leu	MOD	1	1/1 (100.0)	0/1 (0.0)	0/1 (0.0)	0/1 (0.0)	0/1 (0.0)	0/1 (0.0)
10	c.1852T>C	p.Cys618Arg	MOD	8	7/7(100.0)	0/8 (0.0)	4/8 (50.0)	0/8 (0.0)	1/8 (12.5)	0/8 (0.0)
11	c.1900T>C	p.Cys634Arg	H	8	7/8 (87.5)	5/8 (62.5)	5/8 (62.5)	1/8 (12.5)	0/8 (0.0)	0/8 (0.0)
11	c.1901G>A	p.Cys634Tyr	H	1	1/1 (100.0)	1/1 (100.0)	1/1 (100.0)	0/1 (0.0)	0/1 (0.0)	0/1 (0.0)
11	c.1901G>C	p.Cys634Ser	H	1	1/1 (100.0)	0/1 (0.0)	0/1 (0.0)	1/1 (100.0)	0/1 (0.0)	0/1 (0.0)
11	c.1901G>T	p.Cys634Phe	H	5	3/3 (100.0)	1/5 (20.0)	0/5 (0.0)	0/5 (0.0)	0/5 (0.0)	0/5 (0.0)
11	c.1902C>G	p.Cys634Trp	H	14	12/12 (100.0)	2/14 (14.3)	10/14 (71.4)	1/14 (7.1)	0/14 (0.0)	0/14 (0.0)
16	c.2753T>C	p.Met918Thr	HST	1	1/1 (100.0)	0/1 (0.0)	0/1 (0.0)	0/1 (0.0)	0/1 (0.0)	1/1 (100.0)

^a^ In four patients in our series, we only have data about the exon and codon affected, regarding germline RET mutation; they were not included in the table; ^b^ Only carriers with known clinical information were included. Abbreviations: ATA, American Thyroid Association; CLA, cutaneous lichen amyloidosis; HD, Hirschsprung disease; H, high; HPTH, hyperparathyroidism; HST, highest; MEN, multiple endocrine neoplasia, MOD, moderate.

**Table 3 cancers-18-00093-t003:** Classified somatic pathogenic RET variants identified in the study cohort.

Exon	Nucleotide Variant	Aminoacidic Change	No. Carriers ^a^
11	c.1888T>G	p. Cys630Gly	1
11	c.1900T>C	p. Cys634Arg	2
11	c.1901G>A	p. Cys634Tyr	1
16	c.2753T>C	p. Met918Thr	15

^a^ One patient was excluded from this table because we only had data on the affected exon.

## Data Availability

The data presented in this study are available upon request from the corresponding author due to legal and ethical reasons of the home institution.

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
