# Peer review of "Cancers2026, 18(1), 93;https://doi.org/10.3390/cancers18010093"

_cancers, 2025, doi:10.3390/cancers18010093_

Round 1

Reviewer 1 Report

Comments and Suggestions for Authors

This manuscript stands out as an impressive achievement, and the authors should be genuinely proud of the work they have accomplished. Conducting such an extensive, well-designed genetic study in a national cohort is no small task, and your commitment to scientific rigor is evident throughout the paper. By clarifying the specific RET mutation patterns within the Romanian population, you provide crucial data that will directly impact patient care, genetic counseling, and clinical decision-making.

Your findings not only strengthen the global understanding of MTC but also highlight the importance of region-adapted screening strategies — a message that will resonate widely across the endocrinology and oncology community. Moreover, the integration of both germline and somatic testing reflects a forward-looking approach aligned with precision medicine.

Please take pride in knowing that your work fills a significant gap in the literature and will serve as a foundation for future research, improved guidelines, and better outcomes for patients and families. This study is a testament to the power of thorough clinical investigation and collaborative effort — and it undoubtedly sets a strong benchmark for future genetic research in MTC.

Author Response

We thank the reviewer for their valuable time and consideration in reviewing our work. 

Reviewer 2 Report

Comments and Suggestions for Authors

Thank you for the opportunity to review the manuscript “Landscape of Phenotype-Genotype Correlations in Romanian Patients with Medullary Thyroid Carcinoma” by Laura-Semonia Stanescu et al. This is a comprehensive study of a relatively large national cohort of a rare thyroid carcinoma MTC.

Please see my remarks below:

  1. How many relatives were studied? (31 mentioned in the line 108, then it is described as 32)
  2. Line 165: 11/112 (9.8%) patients with apparently sporadic cases were reclassified as hereditary- please provide more clinical and genetic details.
  3. How many patients started the tyrosine kinase inhibitor therapy treatment?

Comments on the Quality of English Language

The English could be improved to more clearly express the research.

Author Response

Comment 1: The English could be improved to more clearly express the research.

Response 1: Yes, the English was improved after another thorough proof-reading. 

Page 2, Lines 64-65: Figure 1 describes the clinical characteristics of hereditary MTC, which includes the four subtypes of MEN2 syndrome and MEN3 syndrome.

Page 4, Line 149: The mention of "2/29 cases" was a typographical error. The correct denominator is 27, as the total number of cases with DNA obtained from tissue is 27 (10 from fresh tissue + 15 from primary FFPE + 2 from metastatic FFPE). We have corrected this to "2/27 cases" in the revised manuscript.

Page 4, Lines 164-164: “To provide a comprehensive overview of the study population, Table 1 outlines the demographic and clinical characteristics of patients with MTC.”

Page 7, Lines 237-239: “Over the past decade, the National Institute of Endocrinology in Bucharest, Romania has established the necessary infrastructure and developed standardized procedures for molecular diagnosis, genetic counselling, and targeted therapeutic interventions in MTC.”

Page 8, Lines 262-266: “The phenotypic characteristics associated with specific genetic alterations have been documented in the literature as follows: for mutations in codon 634, the clinical presentation features MTC in 100% of patients, PHEO in 50% of patients and HPTH in 15% of patients; in addition, around 30% of patients have CLA, known as MEN2 with CLA. Among our patients with mutations in codon 634 etc.”

Page 8, Lines 276-281: We have added in the Discussion section the following limitation: “The disproportionately high fraction could be explained by the selection bias, a limitation of the study. Because our cohort was recruited from a tertiary referral center specializing in endocrine genetics, the patient population was enriched for hereditary cases. This re-cruitment pattern limits the generalizability of our findings, and the observed frequency should therefore be interpreted with caution.”

Page 8, Lines 284-286: “According to published literature, sMTC has been reported to exhibit a markedly higher prevalence of FMTC, accounting for approximately 87%, in contrast to the distribution observed in the classic MEN2 syndrome.”

Page 8, Lines 289-291: “The findings of the present study are consistent with previous reports in the literature, which delineate distinct profiles of RET germline mutations in hereditary medullary thyroid carcinoma compared to cases that appear sporadic. “

Page 8, Line 298: “These observations are consistent with our findings …”

Page 8, Lines 304-305: “Consistent with previously published studies, the M918T mutation emerged as the most prevalent, with a detection rate of 75%”

Page 9, Lines 307-325 and 336-344: We have further refined the discussion of somatic RET mutations by adding an explicit acknowledgment of the study limitation regarding the lack of systematic somatic testing in hereditary cases, while maintaining literature context on rare double-hit.

Comment 2: How many relatives were studied? (31 mentioned in the line 108, then it is described as 32).

Response 2: Thank you for your observation. The correct number of relatives is 32. I have made the corrections accordingly. 

Comment 3: Line 165: 11/112 (9.8%) patients with apparently sporadic cases were reclassified as hereditary- please provide more clinical and genetic details.

 Response 3: Thank you for this comment. While detailed clinical information was not available for all 11 patients, genetic details have already been included in the manuscript. Specifically, germline mutations affecting non‑cysteine residues were more frequent in apparently sporadic cases and V804M mutation was the most prevalent. This information is included in the manuscript at Page 5, Lines 183 and 187-189.

Comment 4: How many patients started the tyrosine kinase inhibitor therapy treatment?

Response 4: We appreciate the reviewer’s question, as it appropriately draws attention to the clinical management and therapeutic implications of somatic RET mutations in our cohort. To enhance clarity and address this comment, we have added the following details to the Results section of the revised manuscript: “In our cohort, 18 patients with sMTC received multikinase tyrosine kinase inhibitor (TKI) therapy, specifically vandetanib or cabozantinib. Among these 18 patients, 14 were found to harbor somatic RET mutations upon testing, with the majority (10/14) carrying the M918T mutation.” Page 7, Lines 230-233.

Reviewer 3 Report

Comments and Suggestions for Authors

Thank you for the manuscript.

Medullary thyroid carcinoma (MTC) is a rare tumour of the thyroid gland. Around 25% of MTC is hereditary and involves germline mutation of the RET proto-oncogene gene. Hereditary MTC most commonly occur as part of MEN 2 and its variants, and rarely MEN 3. Somatic RET mutations are prevalent in patients with sporadic MTC.  Routine testing for at least germline RET mutation is recommended.  However, testing is not available in some countries. Patients with MTC showing RET mutation are amenable to targeted therapy with RET inhibitors. This manuscript reports results of a study that investigated the prevalence of hereditary MTC and landscape of germline and somatic RET mutations in a cohort of patients treated at a centre in Romania. Germline and somatic mutations were analyzed in 164 patients, 59 of whom had hereditary MTC. Year: 2021-2024. 20 clinically diagnosed hereditary MTC, 112 apparent sporadic and 32 first-degree relatives with MTC. Somatic RET in 27 with wild-type but aggressive disease. The Sanger sequencing technique was used. Testing for somatic RET mutation is a requirement for reimbursement for targeted therapy in advanced disease. Genomic DNA from blood and somatic from fresh tissue from primary or metastatic tumour, or FFPE tissue blocks. Germline variant of RET in 35.9% of MTC and highly likely in patients with clinical features of hereditary disease (100%: 20/20) and those with family history: 87.5%: 28/32). Codon 634 most commonly affected (55.9%: 33/59). Common variant in 98% MEN2, and was classical MEN 2 in 72%. Somatic mutation of M 918T was the most prevalent, especially in advanced disease. Majority of patients in the study had either moderate or high-risk germline RET mutations.

Title

Modify the title to align with the aim, introduction, materials and methods and results. The focus was not predominately “Phenotype-Genotype Correlations”.

Suggested title: Germline and somatic RET mutations landscape of medullary thyroid carcinoma in Romania

Simple summary

It is not perfectly aligned. The manuscript is focusing less on targeted therapy. Check and revise “The aim of the research was to coordinate testing for RET mutations with tyrosine kinase inhibitor therapy”. The study focused on the profile of RET mutations in patients with MTC in Romania.

Abstract

No concerns.

Keywords

Consider adding ‘mutations’, ‘germline’, ‘somatic’

Introduction

Add a brief description of Sanger sequencing before a statement on its accuracy as it may not be a familiar topic to some readers. The section is otherwise clear and focused.

Materials and Methods

Study period 2021 – 2024 but ethical clearance 27 June 2022? It is otherwise fine

Statistical Analysis

Check if it is “two-independent t-test” or ‘two-sample independent t-test. Also check data was normally distributed to justify the use to summarize, and thereafter comparison of means.

Results

Check if it is appropriate to use the mean or median in Linec185-187.

Illustrations

Figure 1

No concerns. It is simple, easy to read and appropriate.

Table 1

The appropriateness of using the mean must be included. The mean to be reported with standard deviation. Consider reporting the median. Use n = 112 as the denominator in “Stage MTC at diagnosis” for the percentages to add to 100%. The foot-note is fine.

Table 2

Add the sample size after the caption of Table 2 (n = 42 versus 55) for clarification.

Table 3

No concerns.

Discussion

No concerns.

Limitations

No entry. Consider entering some limitations.

Conclusions

They are concise and based on results from the study.

References

No concerns.

Author Response

Comment 1: Figures and tables can be improved. 

Response 1: All Figure and Tables titled were reworded to be more coherent and provide more understanding. Figure 1 was modified replacing the word variants for the better suited subtype and the arrows were re-adjusted.

Page 2, Line 73: Figure 1 image was modified such that the arrows were re-adjusted. Additionally, the word variant was changed to subtype.

Page 2, Lines 74-75: Figure 1. Characterizing Hereditary MTC in terms of MEN syndromes and their respective pathologies.

Page 4, Line 167. Table 1. Demographic and Clinical Characteristics of Patients with Medullary Thyroid Carcinoma.

Page 6, Lines 206-207: Table 2. Classified Germline RET Variants Identified in the Study Cohort (n = 164; 112 apparently sporadic and 52 hereditary cases).

Page 7, Line 228: Table 3. Classified Somatic Pathogenic RET Variants Identified in the Study Cohort.

Comment 2: Modify the title to align with the aim, introduction, materials and methods and results.

Response 2: We thank the reviewer for this thoughtful suggestion and for highlighting the importance of ensuring the title accurately reflects the content of the manuscript. After careful discussion among the co-authors, we have decided to retain the original title, as we believe it best captures the primary focus and contributions of our study.

Comment 3: Simple summary

Response 3: We thank the reviewer for the constructive observation regarding the alignment of the manuscript’s aim and title. We acknowledge that the initial phrasing placed greater emphasis on coordinating testing with tyrosine kinase inhibitor therapy. Following the reviewer’s advice, we have made minor revisions to better reflect the actual aim of the study, which focuses on the mutational profile of RET in Romanian patients with medullary thyroid carcinoma.

The revised aim: "The aim of the research was to characterize RET mutations in patients with medullary thyroid carcinoma (MTC) in Romania, with consideration of their relevance for tyrosine kinase inhibitor therapy." Page 1, Lines 12-14

Slight rephrasing for smoother flow (“performed testing” instead of “done testing”). Page 1, Line 15

Comment 4: Keywords

Response 4: We thank the reviewer for the helpful suggestion regarding the keywords. In line with this advice, we have revised the list to better reflect the scope of the study. The updated keywords are: "medullary thyroid cancer; RET; sporadic; hereditary; mutations; germline; somatic". Page 1, Lines 37-38

Comment 5: Materials and Methods

Response 5: We thank the reviewer for noting the apparent discrepancy between the study period and the date of ethical clearance. To clarify, while the overall study period was 2021–2024, patient recruitment and data collection commenced only after ethical approval was obtained on 27 June 2022. The earlier date reflects preparatory work and retrospective identification of eligible cases, whereas all procedures involving patient data were conducted in accordance with the approved protocol. We have revised the Materials and Methods section to make this distinction clearer.

The revised Materials and Methods: "The study was conducted between 2021 and 2024, with patient recruitment and data collection commencing only after ethical approval was obtained from the Ethics Committee and Scientific Council of the “C.I. Parhon” National Institute of Endocrinology, Bucharest, Romania (approval no. 12, 27 June 2022)." Page 3, Lines 117-120

Moreover, strengthens coherence “defined by” instead of “characterized by”. Page 3, Line 123

Comment 6: Statistical Analysis

Response 6:  We thank the reviewer for pointing out that the spelling mistake “Two‑independent t‑test” instead of the correct and standard term “Two‑tailed independent t‑test”. As evidenced by the fact that further in the manuscript when the mean ages were compared between germline RET variants and wild-type patients, we state the analysis is two-tailed independent t-test along with the p-value, t-value and alpha (Page 6, Line 191-193). Furthermore, the dataset did undergo normality assessment otherwise the two‑tailed independent t‑test could not have been used.

Page 4, Line 154-157: A two-tail independent t-test (alpha=0.05) was performed to determine if there was a significant difference in the mean ages between germline RET variant patients (59) and wild-type patients (105).

Comment 7: Results

Response 7: We thank the reviewer for the observation. In this section we are reporting categorical prevalence data (proportions of RET mutations in already known syndromic cases compared with apparently sporadic cases), analyzed using chi‑square testing. As these are categorical variables, measures of central tendency such as mean or median are not applicable. 

Comment 8: Table 1 

Response 8: We thank the reviewer for the valuable comments. In Table 1, we have revised the age at diagnosis to be reported as mean ± standard deviation and additionally provided the median and interquartile range to account for distribution. We have also recalculated the percentages for “Stage MTC at diagnosis” using n = 112 as the denominator, ensuring that the values sum to 100%. The footnote has been retained as originally written.

Comment 9: Table 2 

Response 9:  We thank the reviewer for this helpful suggestion. Germline RET testing was performed in 164 subjects, including 112 apparently sporadic cases and 52 hereditary cases. To improve clarity, we have added the sample size information directly to the caption of Table 2: Table 2. Classified Germline RET Variants Identified in the Study Cohort (n = 164; 112 apparently sporadic and 52 hereditary cases).”  Page 6, Lines 206-207

Comment 10: Limitation

Response 10: We have added in the Discussion section the following limitations:

  • selection bias, as a possible explanation for the high percentage of reclassified MTC patients from sporadic to hereditary. Page 8, Lines 277-282.
  • lack of systematic somatic testing in hereditary cases, while maintaining literature context on rare double-hit events Page 9, Lines 341-344
